# Single-Taper Conical Tapered Stem in Total Hip Arthroplasty for Primary Osteoarthritis: A Comparative Long-Term Registry Evaluation

**DOI:** 10.3390/jcm13195943

**Published:** 2024-10-06

**Authors:** Francesco Castagnini, Barbara Bordini, Monica Cosentino, Mara Gorgone, Andrea Minerba, Marco Rotini, Emanuele Diquattro, Francesco Traina

**Affiliations:** 1SC Ortopedia-Traumatologia e Chirurgia Protesica e dei Reimpianti di Anca e Ginocchio, IRCCS Istituto Ortopedico Rizzoli, via Pupilli 1, 40136 Bologna, Italy; mara.gorgone@ior.it (M.G.); andrea.minerba@ior.it (A.M.); marco.rotini@ior.it (M.R.); emanuele.diquattro@ior.it (E.D.); francesco.traina@ior.it (F.T.); 2Laboratorio di Tecnologia Medica, IRCCS Istituto Ortopedico Rizzoli, via di Barbiano 1/13, 40136 Bologna, Italy; barbara.bordini@ior.it (B.B.); monica.cosentino@ior.it (M.C.); 3Dipartimento di Scienze Biomediche e Neuromotorie DIBINEM, University of Bologna, via Irnerio 48, 40126 Bologna, Italy

**Keywords:** tapered, conical, anteversion, antetorsion, aseptic loosening, Wagner cone, osteoarthritis

## Abstract

**Background/Objectives**: Single-taper conical tapered stems (STCTSs) are possible options for femoral reconstruction in THA performed for primary osteoarthritis, but outcomes are poorly ascertained. A specific STCTS in THA performed for primary osteoarthritis was investigated and compared to a control group including all the other cementless stems, aiming to assess the following: (1) the survival rates of the two cohorts and the hazard ratios for failure; (2) the survival rates and the hazard ratios for failures for stem failure, stem aseptic loosening, and periprosthetic fracture. **Methods**: A regional arthroplasty registry study evaluating a specific STCTS in THA performed for primary osteoarthritis was designed. A control group including all the other cementless stems was considered. **Results**: In total, 1773 STCTSs were compared to 37,944 cementless stems. The cumulative survivorship of the STCTS cohort was 95.6% at 10 years and 92.7% at 15 years, which was not different to the control group (*p* = 0.252). After age stratification, the hazard ratio for STCTS failure was not different to the control group. With stem revision as the endpoint, the STCTS cohort outperformed the control group (at 10 years, 98% versus 96.8%; *p* < 0.001). The STCTSs achieved better survival rates in females <65 years (*p* = 0.023). With stem aseptic loosening as the endpoint, the survival rates did not differ between the two cohorts (*p* = 0.085), as well as the adjusted hazard ratios (*p* = 0.264). With periprosthetic fracture as the endpoint, the survival rates were better for the STCTSs (*p* < 0.001). **Conclusions**: STCTSs in THA for primary osteoarthritis provided dependable long-term outcomes, not inferior to all the other cementless stems with various designs.

## 1. Introduction

Single-taper conical tapered stems (STCTSs) in THA were designed to overcome some typical reconstructive challenges of developmental dysplasia of the hip, namely the high neck-shaft angle, low offset, narrow femoral canal, proximal–distal mismatch, and high antetorsion angles [1,2,3,4]. Specifically, conical tapered stems were considered appropriate in the case of femoral antetorsions higher than 25° and in the case of distorted axial and coronal morphologies, in particular with narrow femoral canals. In the case of femoral antetorsion ≥25°, the reconstruction of three-dimensional biomechanical parameters provided by an STCTS was considered acceptable in 88% of the dysplastic cases, an outcome significantly better than that provided by anatomical or flat wedge tapered stem designs [5,6]. On the other hand, STCTSs achieved low performances in femoral offset reconstruction when the native femoral offset in dysplastic hips was higher than the normal thresholds [6,7]. Another relevant drawback of STCTSs was the diaphyseal fixation, with violation of the distal bone stock and possible higher grade femoral bone loss in the case of femoral revision [6,7].

STCTSs demonstrated dependable long-term outcomes in dysplastic hips, with modest subsidence rates in the first three months (18%), non-negligible proximal stress shielding (28%), and very high rates of osseointegration (78–100%) [7,8,9,10,11,12,13,14]. Since the positive results of these stems in developmental dysplasia of the hip, STCTSs were also adopted in exceptional cases in which a cementless flat wedge stem or a cemented stem were not indicated, namely small femurs, excessive antetorsion, low offset, poor metaphyseal bone quality, and complex anatomies in general [7,8,9,10,11,12,13,14]. The main retrospective case series reported good-to-excellent outcomes, with valid osseointegration (98–100%) and mid-term survival rates higher than 95% [7,8,9,10,11,12,13,14].

However, the use of an STCTS in THA performed for primary osteoarthritis in a sizeable population is limited and there is a substantial lack of comparative long-term studies [15,16,17,18]. Thus, an arthroplasty registry study was designed, aiming to assess and compare the survival rates of a specific STCTS in THA performed for primary osteoarthritis to a control group of cementless stems with different stem designs. The aims of this paper were to assess and compare (1) the survival rates of the two cohorts and the hazard ratios for failure, and (2) the survival rates and the hazard ratios for failures for specific endpoints (stem failure, stem aseptic loosening, and periprosthetic fracture). We hypothesized that a specific STCTS in THA performed for primary osteoarthritis could achieve long-term survival rates comparable to cementless stems with different designs, even when considering specific endpoints.

## 2. Materials and Methods

Institutional review board approval was not required for this study due to the standard practice of data anonymization and registry data collection.

The RIPO (Registro dell’Implantologia Protesica Ortopedica) registry has been documenting hip, knee, and shoulder arthroplasty procedures (primary and revision) since January 2000 [19,20,21]. This registry covers around 4.5 million individuals and involves 69 orthopedic facilities in the Emilia-Romagna region [19,20,21]. Surgeons must complete detailed forms for each primary and revision procedure, including patient demographics, implant details, and surgical techniques [19,20,21]. The collected data are crosschecked with other sources and databases; a 98% capture rate is achieved, similar to the most important national registries [19,20,21]. The remaining 2% of data are missing due to lack of adherence [19,20,21]. The RIPO registry is a member of the ISAR—the International Society of the Arthroplasty Registries [19,20,21].

The RIPO registry was used to identify THA performed for primary hip osteoarthritis and with a specific STCTS with splines, Wagner Cone, or Conus (Zimmer, Warsaw, IN, USA) between 2000 and 2020. A control group with all cementless stems in THA performed for primary osteoarthritis in the same time span was selected. The inclusion criteria were as follows: single-taper stem design, primary osteoarthritis, and cementless fixation. Only residing patients were included in order to minimize the bias to loss to follow-up. The exclusion criteria were as follows: dual mobility implants, metal-on-metal THA with head size ≥ 36 mm, resurfacing hip arthroplasty, and modular implants.

The cohort with the index stem encompassed 1773 implants, and 37,944 THAs in the control group. The demographic and implant features of the two cohorts are described and compared in Table 1.

The survival rates of the two cohorts, as well as the hazard ratios for failure, were compared; specific endpoints (stem aseptic loosening, periprosthetic infection, and periprosthetic fracture) were also considered.

The demographics and implant features were different in the two cohorts: sex (*p* = 0.022), mean age at implant (*p* < 0.001), age decades at implant (*p* < 0.001), age per group (*p* < 0.001), bearing surfaces (*p* < 0.001), and head size (*p* < 0.001) [Table 1].

In the index stem cohort, the five most common implants were Fitmore Zimmer/Wagner Cone (793, 44.7%), CLS Zimmer/Wagner Cone (308, 17.4%), Continuum Zimmer/Wagner Cone (182, 10.3%), Standard Cup Zimmer/Wagner Cone (74, 4.2%), and Trilogy Zimmer/Wagner Cone (66, 3.7%). In the control group, the five most common implants were Fixa Ti-Por/Corae (Adler Ortho, Milan, Italy) (1442, 3.8%), ABGII/ABGII (Stryker, Kalamazoo, MI, USA) (1431, 3.8%), EP-Fit Plus/SL Plus (Smith and Nephew, London, UK) (1403, 3.7%), R3/Polarstem (Smith and Nephew, London, UK) (1340, 3.5%), and R3/SL Plus MIA (Smith and Nephew, London, UK) (1296, 3.4%).

The median follow-up period was 10.9 years for the index stem cohort (IQR: 6.6–15.3) and 5.7 years for the control group (IQR: 2.7–10.2).

The survival rates (endpoint: overall failures) of the two cohorts were assessed and compared, and were stratified for age and sex. Some of the most implanted stems with different designs from the control group were selected and compared using a multivariate analysis, evaluating the adjusted hazard ratios for stem failure and for stem aseptic loosening. The hazard ratios for failure stratified for age group were reported. Reasons for revision were listed. The same methodology was adopted (when feasible) for specific endpoints, namely stem failure, stem aseptic loosening, and periprosthetic fracture.

### 2.1. Index Stem Features

Wagner Cone is a cementless STCTS introduced on the market in 1992 that was slightly modified in 2006 [7,14,22]. The titanium–aluminum–niobium alloy (PROTASUL-100) stem has a tapered angle of 5°, with eight longitudinal sharp ribs; this design allows for an independent tuning of the femoral antetorsion, while providing proximal diaphyseal fixation and rotational stability [7,14,22]. The proximal biomechanics are designed for dysplastic cases, with reduced femoral offset and two neck-shaft angles—125° and 135°; for both the versions, 12 diameters (from 13 to 24 mm) are available [7,14,22] (Figure 1).

### 2.2. Statistical Analysis

Continuous data were expressed with a median and interquartile range, not following a Gaussian distribution. The qualitative data were expressed as frequencies and percentages. Demographic- and implant-related data were analyzed using Wilcoxon rank sum test and Pearson’s Chi-squared test. The choice of statistical analyses was based on the guidelines by Ranstam et al. [23]. The survivorship of the THA implants was calculated and plotted according to the Kaplan–Meier method, where each curve was flanked by a pair of curves, indicating a 95% confidence interval. The endpoint was the removal or change of any component; specific endpoints were also considered (stem failure, stem aseptic loosening, and periprosthetic fracture). The implants were followed until the last date of observation (date of death or 31 December 2020). The Log-Rank test was used to compare survival curves between the groups. The Cox multiple regression model for analyzing survivorship was considered. The proportionality hazards assumption was tested by the Schoenfeld residual method. Statistical analysis was performed using R Core Team 4.3.0 (2023). A *p*-value less than 0.05 was considered statistically significant.

## 3. Results

The cumulative survivorship of the STCTS cohort was 95.6% at 10 years (95%CI: 94.6–96.7%, 998 exposed implants) and 92.7% at 15 years (95%CI: 91.1–94.3%, 472 exposed implants). No significant differences were noted when compared to the control group (*p* = 0.252, Log-rank test); at 10 years, the survival rate was 94.8% (95%CI: 94.5–95.1%, 9793 exposed implants) and at 15 years, it was 91.7% (95%CI: 91.2–92.2%, 3360 exposed implants) (Figure 2).

The reasons for revision are listed in Table 2.

Some of the most implanted stems from the control group were selected and compared to the STCTS cohort using a multivariate analysis—ADR Smith and Nephew (single-taper conical stem); CLS Zimmer (single-taper single-wedge tapered stem); Corail Depuy (Warsaw, IN, USA: single-taper single-wedge tapered stem); Polarstem Smith and Nephew (single-taper single-wedge tapered stem); Sl-Plus Smith and Nephew (single-taper tapered rectangle stem); and Taperloc Biomet (Warsaw, IN, USA: single-taper single-wedge tapered stem). The adjusted hazard ratios for stem failure were as follows: 2.18 (CI95%: 1.14–4.18; *p* = 0.019) for ADR; 1.54 (CI95%: 1.04–2.26; *p* = 0.03) for CLS; 0.99 (CI95%: 0.53–1.85; *p* = 0.968) for Corail; 0.65 (CI95%: 0.33–1.28; *p* = 0.216) for Polarstem; 2.31 (CI95%: 1.58–3.38; *p* < 0.001) for Sl-Plus; 0.62 (CI95%: 0.36–1.05; *p* = 0.073) for Taperloc. The adjusted hazard ratios for stem aseptic loosening were as follows: 3.66 (CI95%: 1.46–9.18; *p* = 0.006) for ADR; 1.17 (CI95%: 0.56–2.45; *p* = 0.683) for CLS; 1.05 (CI95%: 0.36–3.05; *p* = 0.929) for Corail; 0.54 (CI95%: 0.17–1.76; *p* = 0.305) for Polarstem; 4.17 (CI95%: 2.19–7.97; *p* < 0.001) for Sl-Plus; 0.57 (CI95%: 0.22–1.43; *p* = 0.228) for Taperloc. 

When the population was stratified for age and sex, the two cohorts showed similar survival rates (Figure 3).

The hazard ratio for STCTS failure was 0.91 (95%CI: 0.67–1.22, *p* = 0.509) in the population aged less than 65, and 0.83 (95%CI: 0.62–1.10, *p* = 0.195) in the population aged 65 or older.

Considering stem revision as the endpoint, the cumulative survivorship of the index stem cohort was 98% (95%CI: 97.3–98.8%) at 10 years versus 96.8% (95%CI: 96.5–97%) for the control group, and 97.5% (95%CI: 96.6–98.3%) at 15 years versus 94.8% (95%CI: 94.4–95.3%) for the control group (*p* < 0.001, Log-rank test) (Figure 3). When the population was stratified for age and sex, the two cohorts showed similar survival rates for stem revision, apart from in females aged less than 65 (where STCTS performed better: *p* = 0.023, Log-rank test).

The hazard ratio adjusted for sex for index stem revision was 0.47 (95%CI: 0.27–0.80, *p* = 0.005) in the population aged less than 65.

The distribution of the stem failures according to the reason for revision and time of onset is depicted in Figure 4 for both the cohorts.

When the endpoint was stem aseptic loosening, the survival rates did not differ between the two cohorts (*p* = 0.085, Log-rank test). At 10 years, the survival rates were 99.3% (95%CI: 98.9–99.7%) for the index stem cohort and 99% (95%CI: 98.8–99.1%) for the control group. At 15 years, the survival rates were 99.3% (95%CI: 98.9–99.7%) for the index stem cohort and 98.6% (95%CI: 98.4–98.8%) for the control group.

When the population was stratified for age and sex, the two cohorts showed similar survival rates for stem aseptic failures.

The hazard ratio adjusted for sex for stem revision for the index stem was 0.63 (95%CI: 0.28–1.42, *p* = 0.264) in the population aged 65 or older.

When the endpoint was periprosthetic fracture, the survival rates differed between the two cohorts (*p* < 0.001, Log-rank test). At 10 years, the survival rates were 100% (95%CI: 100–100%) for the index stem cohort and 99.1% (95%CI: 99–99.3%) for the control group. At 15 years, the survival rates were 100% (95%CI: 100–100%) for the index stem cohort and 98.2% (95%CI: 97.9–98.5%) for the control group.

When the population was stratified for age and sex, the index stem cohort had lower rates of periprosthetic fractures in females aged 65 or more (*p* = 0.002, Log-rank test).

## 4. Discussion

STCTSs in THA performed for primary osteoarthritis demonstrated dependable long-term survival rates at 10 and 15 years, with non-inferior outcomes compared to all the other cementless implants. When stem revision was the endpoint, the index stem showed even higher long-term survival rates, in particular in females aged less than 65. The STCTS showed a negligible rate of failures for periprosthetic fracture, lower than all the other cementless stems, especially in females aged 65 or more. The hypothesis that the index stem could achieve non-inferior survival rates in the long term was confirmed.

The main strength of this paper is the very sizeable population using a specific STCTS in THA for primary osteoarthritis, which was compared to a large control group of other cementless stems with multiple designs. The limitations of this paper are related to the registry nature of the study. First, the definition of primary osteoarthritis can be generic, not shareable, and sometimes misleading, as it was given by different surgeons performing THA; the possible wide spectrum of the femoral morphologies that were treated with the index stem could not be assessed. Clinical outcomes were not evaluated; thus, it is not possible to define if the STCTS is protective towards thigh pain, as suggested in the literature [7,22]. Radiological analysis was not performed; thus, it is not possible to profile the ideal candidates for the index stems on the basis of pre-operative radiographs and morphologies. This is a remarkable drawback, as this paper does not enlighten if other femoral components with more proximal fixation and less distal bone violation may be sufficient to address at least some morphologies [7,14,15]. Moreover, it is not possible to ascertain the post-operative radiographic outcomes that are still debated in the literature, namely subsidence rates, subsidence magnitude, and proximal stress shielding [7,8,9,10,11,12,13,14,15,16,17,18]. Furthermore, the registry study cannot capture complications not leading to component exchange, such as conservatively treated dislocations, periprosthetic fractures with no stem revision, and low-performing implants not doomed to revision [19,20,21].

The STCTS achieved a 10-year survival rate higher than 95%, on par with the most recent benchmark for THA and with all the other cementless stems of the control group [24,25,26,27]. The most sizeable case series about the same stem adopted in dysplasia or difficult anatomies favorably compared with the present report, even if the longest mean follow-up in the literature is less than 10 years [14,15,16,17,18].

Even the rate of stem revisions was very modest, similar to the other case series reporting 98% at short-to-mid-term follow-ups [14,15,16,17,18]. In the present report, STCTSs outperformed the control group and showed that patients aged 65 or less, in particular females, were good candidates for the STCTS; this finding is unedited and tends to support the long-term positive outcomes even in younger patients with primary osteoarthritis. Moreover, STCTSs were demonstrated to be a good option even when compared to other common stems; they were superior to another conical stem design (ADR) and the tapered rectangle stem design (Sl-Plus) in terms of stem failures and stem aseptic loosening.

Stem aseptic loosening occurred with similar percentages among the index stem cohort and control group (99.3% vs. 99% at 10 years). Similar rates were also reported in the main case series of STCTSs (98.6–100% at short-to-mid-term follow-ups) [9,10,11,12]. The authors agreed that conical tapered stems provided a solid fixation with valid osseointegration in (almost) all the cases, even in complex femoral morphologies; this finding might also be indirectly confirmed in the present report, in the long term [7,8,9,10,11,12,13,14,15,16,17,18].

Surprisingly, no periprosthetic fracture requiring stem revision occurred in the index stem cohort; with this endpoint, STCTSs outperformed the control group. This finding needs further explorations; it may be due to diaphyseal secure fixation, which may hinder loosening, even due to trauma. However, this report did not highlight if periprosthetic fractures not requiring stem exchange might be more frequent, or even more severe in STCTSs.

## 5. Conclusions

In a sizeable registry population, STCTSs in THA for primary osteoarthritis provided dependable long-term outcomes, not inferior to all the other cementless stems with different designs. The rate of stem loosening was even lower than the control group, in particular in females aged less than 65 years, as well as the rate of periprosthetic fractures. However, this report did not profile the ideal candidate for this stem. As it is difficult to claim for a randomized controlled trial, a rule of thumb can be inferred from the current literature and the present case series: considering that STCTSs require a proximal diaphyseal fixation with distal bone stock violation, it should be concluded that the most suitable conditions for this stem are complex anatomies with compromised proximal metaphyseal fixation, excessive antetorsion, and narrow femoral canals. Even in these cases, valid long-term outcomes of STCTSs should be expected.

## Figures and Tables

**Figure 1 jcm-13-05943-f001:**
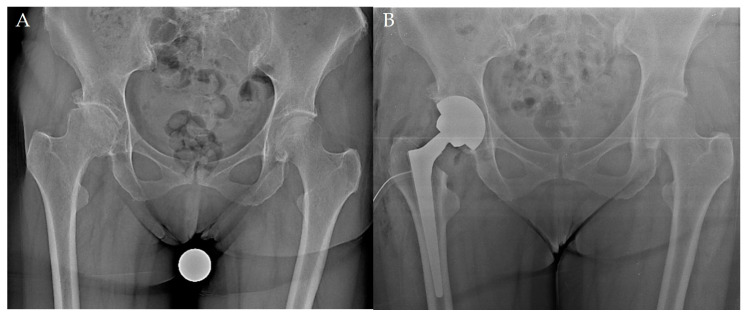
The post-operative radiograph of a Wagner Cone stem (**B**) implanted in a 52-year-old female with primary hip osteoarthritis (**A**).

**Figure 2 jcm-13-05943-f002:**
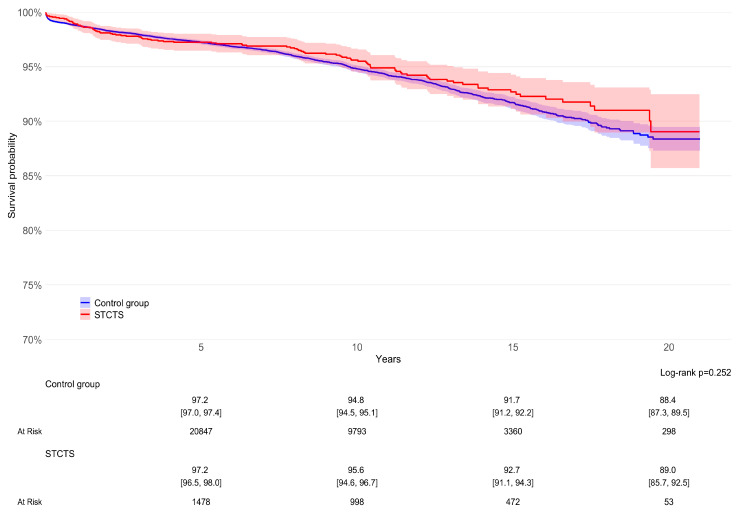
The survival rates of the index stem cohort (in red) and the control groups (in blue) were similar in the long term (*p* = 0.252).

**Figure 3 jcm-13-05943-f003:**
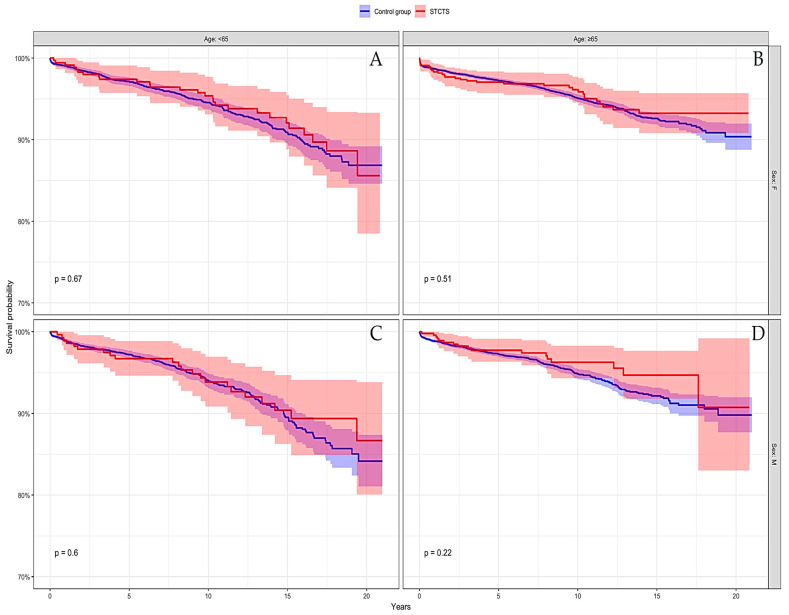
When the population was stratified for age and sex, the two cohorts showed similar survival rates for STCTS (in red) and the control group (in blue). (**A**) Females aged less than 65; (**B**) females aged 65 or older; (**C**) males aged less than 65; (**D**) males aged 65 or older.

**Figure 4 jcm-13-05943-f004:**
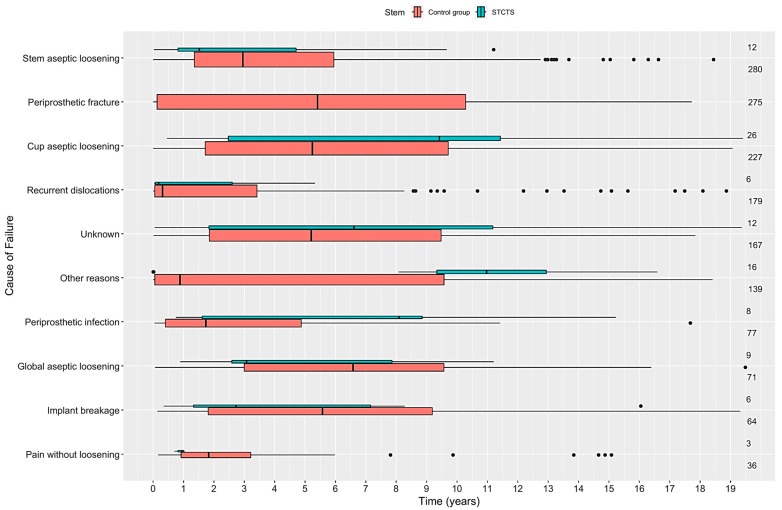
The distribution of the failures according to the time of onset; STCTS in green and control group in red.

**Table 1 jcm-13-05943-t001:** Demographic- and implant-related features showed that the two cohorts differed for sex (*p* = 0.022), mean age at implant (*p* < 0.001), age decades at implant (*p* < 0.001), age per group (*p* < 0.001), bearing surfaces (*p* < 0.001), and head size (*p* < 0.001).

Descriptive Statistic	Index Stem	Control Group	*p* Value
N° of implants	1773	37,944	
Sex Female (%) Male (%)	1015 (57.2%) 758 (42.8%)	20,660 (54.4%) 17,284 (45.6%)	0.022
Median age at implant in years (IQR)	68 (61–73)	70 (64–76)	<0.001
Age per decades <40 (%) 40–49 (%) 50–59 (%) 60–69 (%) 70–79 (%) ≥80	7 (0.4%) 85 (4.8%) 276 (15.6%) 661 (37.3%) 632 (35.6%) 112 (6.3%)	175 (0.5%) 1208 (3.2%) 4511 (11.9%) 11,702 (30.8%) 15,529 (40.9%) 4819 (12.7%)	<0.001
Age per group ≤65 years (%) >65 years (%)	637 (35.9%) 1136 (64.1%)	10,649 (28.1%) 27,295 (71.9%)	<0.001
BMI Underweight (%) Normal weight (%) Overweight (%) Obese (%) Unknown (%)	8 (0.5%) 443 (28.5%) 712 (45.8%) 390 (25.1%) 220	196 (0.6%) 8628 (26.7%) 15,191 (47%) 8332 (25.8%) 5597	0.434
Bearing surfaces Delta-on-X-linked polyethylene (%) Delta-on-Delta (%) Forte-on-polyethylene (%) Metal-on-polyethylene (%) Metal-on-metal (%) Others—unknown (%)	367 (26.4%) 126 (7.1%) 261 (14.8%) 401 (22.7%) 180 (10.2%) 153	6040 (16%) 13,314 (35.2%) 2308 (6.1%) 2832 (7.5%) 810 (2.1%) 9751	<0.001
Head size ≤28 mm (%) 32 mm (%) ≥36 mm (%) Unknown (%)	1231 (69.5%) 148 (8.4%) 391 (22.1%) 3	10,255 (27%) 10,727 (28.3%) 16,934 (44.7%) 28	<0.001

**Table 2 jcm-13-05943-t002:** Reasons for revision of the two cohorts; among implant breakages, there was one stem failure for each cohort.

Reasons for Revision	Index Stem Cohort			Control Group		
	Incidence (out of 1773 Cases)	Percentage (%)	Distribution of the Failures (%)	Incidence (out of 37,944 Cases)	Percentage (%)	Distribution of the Failures (%)
Recurrent dislocations	6	0.3	6.1	179	0.5	11.8
Cup aseptic loosening	26	1.5	26.5	227	0.6	15
Periprosthetic infection	8	0.5	8.2	77	0.2	5.1
Periprosthetic fracture	0	0.6	0	275	0.7	18.2
Stem aseptic loosening	12	0.7	12.2	280	0.7	18.5
Global aseptic loosening	9	0.5	9.2	71	0.2	4.7
Pain without loosening	3	0.2	3.1	36	0.1	2.4
Implant breakage	6	0.3	6.1	64	0.2	4.2
Wear	10	0.6	10.2	38	0.1	2.5
Unknown	12	0.7	12.2	167	0.4	11
Other reasons	6	0.3	6.1	101	0.3	6.7
TOTAL	98	5.5	100	1515	4.0	100

## Data Availability

Data are contained within the article.

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
