# Peer review of "Single-Taper Conical Tapered Stem in Total Hip Arthroplasty for Primary Osteoarthritis: A Comparative Long-Term Registry Evaluation"

_jcm, 2024, doi:10.3390/jcm13195943_

Round 1

Reviewer 1 Report

Comments and Suggestions for Authors

Abstract Comments and Suggestions:

In the abstract, authors should provide sufficient information about the study design, sample size, or statistical methods used that are critical to assessing the validity and reliability of the results. The authors draw general conclusions about the reliability of the STCTS without emphasizing potential limitations or specific contexts in which it may not work as well. The authors should also emphasize how their results impact clinical practice and patient care. The abstract could benefit from a brief explanation of why this study is essential to orthopedics, particularly concerning the challenges associated with primary osteoarthritis.

Manuscript Comments and Suggestions:

In the introduction, the authors mention challenges in DDH; however, they do not address other potential patient populations or scenarios in which the STCTS may be beneficial. The study hypothesis should be clear to provide a focused framework for the study.

The study design should include randomization, which may introduce selection bias and affect the validity of the results.

The study mentions that institutional review board approval was not required; however, some ethical issues remain, such as informed consent and data privacy.

Details regarding the statistical analysis methods, such as how confounding variables were controlled or sample size calculations were performed, are lacking. More detailed elaboration of the statistical analysis methods would be needed.

The clinical implications of the findings in the results section would clarify this aspect.

The discussion acknowledges some limitations; however, potential biases and confounders are not well presented.

The manuscript appears to make broad generalizations about the superiority of STCTS without adequately considering variability in patient populations or surgical techniques.

The conclusion could be more specific regarding the study's limitations and the exact contexts in which STCTS is most beneficial.

The manuscript does not adequately address broader implications, such as cost-effectiveness or potential challenges in the widespread adoption of STCTS.

The manuscript presents a valuable study of the long-term outcomes of single-taper tapered stems in total hip arthroplasty for primary osteoarthritis.

The study objectives are to compare the survival rates of a specific single-taper tapered stem (STCTS) with a control group of other cementless stems in total hip arthroplasty (THA) for primary osteoarthritis.

While the introduction outlines the challenges associated with using STCTS, it could have provided more detail explaining the need for this study and how it fills gaps in existing research.

The exclusion criteria could introduce selection bias, mainly the exclusion of dual-mobility and other specific types of implants. It would be beneficial to discuss how this may affect the results.

The methodology does not account for potential confounding factors that could affect the results, such as comorbidities or differences in surgical techniques. The results are well organized and presented with precise statistical analysis, including survival rates and hazard ratios.

While statistical significance is necessary, the practical relevance of the findings, especially concerning clinical implications, could have been discussed in more detail.

The discussion effectively interprets the results by comparing them with the existing literature and emphasizing the study's contribution to the field.

Although the discussion mentions the results, it does not delve into the underlying mechanisms that could explain them.

There appears to be a slight bias in favor of STCTS, as negative aspects or potential drawbacks are not as thoroughly studied as positive results.

The manuscript generally provides practical insights for clinicians regarding using STCTS in specific patient populations.

Although the conclusion provides insights, it lacks specific recommendations for clinical practice or further research.

Using an extensive registry data set increases the robustness and generalizability of the findings.

The study provides a detailed analysis of survival rates and hazard ratios, offering valuable insights into the efficacy of STCTS in hip arthroplasty.

The manuscript would benefit from more context regarding the clinical implications of the findings, mainly how they translate to patient care.

There is a tendency to favor positive STCTS outcomes without carefully examining potential drawbacks or adverse outcomes.

The study did not adequately account for potential confounding variables that could affect outcomes, such as patient comorbidities or variations in surgical techniques.

The study focuses primarily on quantitative data, with limited exploration of patient-centered outcomes such as quality of life or satisfaction.

Future research directions or unanswered questions should be developed to guide further research in this area.

Comments on the Quality of English Language

Minor grammatical errors throughout the text can be corrected to improve readability.

Author Response

REVIEWER 1

In the abstract, authors should provide sufficient information about the study design, sample size, or statistical methods used that are critical to assessing the validity and reliability of the results. The authors draw general conclusions about the reliability of the STCTS without emphasizing potential limitations or specific contexts in which it may not work as well. The authors should also emphasize how their results impact clinical practice and patient care. The abstract could benefit from a brief explanation of why this study is essential to orthopedics, particularly concerning the challenges associated with primary osteoarthritis.

The abstract can be 250 words and now it is 250 words, and most of them are results. Considering that we cannot remove results, we revised the intro and added the sample size, which were the most relevant omissions. We cannot revise more. All the other concerns should be detailed in the main text.

Changes: line 25: Single-taper conical tapered stems (STCTS) are possible options for femoral reconstruction in THAs performed for primary osteoarthritis, but outcomes are poorly ascertained. Line 32: 1773 STCTS were compared to 37944 cementless stems

Manuscript Comments and Suggestions:

In the introduction, the authors mention challenges in DDH; however, they do not address other potential patient populations or scenarios in which the STCTS may be beneficial. The study hypothesis should be clear to provide a focused framework for the study.

Wagner Cone was designed specifically for dysplastic hips. This was written and explained by the creator, Wagner. Then the stem was so successful that it was adopted in other scenarios, mostly complex anatomies, but also primary osteoarthritis in some morphological outliers.

Please read again line 64-72. You will find the explanations and the details.

Hypothesis also given.

The study design should include randomization, which may introduce selection bias and affect the validity of the results.

The randomization is not adopted in arthroplasty registry studies. It is not possible. And it is quite useless, as the sample sizes are very high and, consequently, tend to randomization and regression towards the mean.

The study mentions that institutional review board approval was not required; however, some ethical issues remain, such as informed consent and data privacy.

There are no ethical issues. It is a common practice for every national arthroplasty registry. Moreover, RIPO adheres to ISAR and to international standard practice for arthroplasty registries. Line 87: Institutional board approval was not required for this study due to the standard practice of data anonymization and registry data collection.

Details regarding the statistical analysis methods, such as how confounding variables were controlled or sample size calculations were performed, are lacking. More detailed elaboration of the statistical analysis methods would be needed.

This is all the statistical analysis. It is complete.

The quantitative data were detailed as average values, standard deviations and ranges of minimum and maximum. The qualitative data were expressed ss frequencies and percentages. Demographic and implant related data were analyzed using Welch Two sample t-test and Pearson’s Chi-squared test. The survivorship of the THA implants was calculated and plotted according to Kaplan-Meier method: each curve was flanked by a pair of curves indicating 95% confidence interval. The endpoint was removal or change of any component: specific endpoints were also considered (stem failure, stem aseptic loos-ening, periprosthetic fracture). The implants were followed until the last date of observation (date of death or 31 December 2020). The Log-Rank test was used to compare survival curves between the groups. The Cox multiple regression model for analyzing survivorship was considered. The proportionality hazards assumption was tested by the Schoenfeld re-sidual method. The statistical analysis was performed using R Core Team (2023). A p-value less than 0.05 was considered statistically significant.

The clinical implications of the findings in the results section would clarify this aspect.

Clinical implications cannot be included in registry study

The discussion acknowledges some limitations; however, potential biases and confounders are not well presented.

There is a dedicated section. Line 217-235

The manuscript appears to make broad generalizations about the superiority of STCTS without adequately considering variability in patient populations or surgical techniques.

Out of the study design and purpose

The conclusion could be more specific regarding the study's limitations and the exact contexts in which STCTS is most beneficial.

The manuscript does not adequately address broader implications, such as cost-effectiveness or potential challenges in the widespread adoption of STCTS.

The two together. This is the conclusion.

“As it is difficult to claim for some randomized controlled trial, a rule of thumb can be inferred from the current amount of literature: considering that STCTS requires a proximal diaphyseal fixation with distal bone stock violation, it should be concluded that the most suitable conditions for this stem are complex anatomies with compromised proximal metaphyseal fixation, excessive anteversion and narrow femoral canals. Even in these cases, valid long-term outcomes of STCTS should be expected.”

Cost-effectiveness cannot be assessed. Widespread adoption is discouraged in the conclusion section.

The manuscript presents a valuable study of the long-term outcomes of single-taper tapered stems in total hip arthroplasty for primary osteoarthritis.

The study objectives are to compare the survival rates of a specific single-taper tapered stem (STCTS) with a control group of other cementless stems in total hip arthroplasty (THA) for primary osteoarthritis. You get it While the introduction outlines the challenges associated with using STCTS, it could have provided more detail explaining the need for this study and how it fills gaps in existing research. Provided in the manuscript: almost ten lines. The exclusion criteria could introduce selection bias, mainly the exclusion of dual-mobility and other specific types of implants. It would be beneficial to discuss how this may affect the results. No, on the contrary the exclusion criteria were adopted to reduce confounding variables. The methodology does not account for potential confounding factors that could affect the results, such as comorbidities or differences in surgical techniques. It is a registry study: how can you do it? The results are well organized and presented with precise statistical analysis, including survival rates and hazard ratios. While statistical significance is necessary, the practical relevance of the findings, especially concerning clinical implications, could have been discussed in more detail. Done: see above. The discussion effectively interprets the results by comparing them with the existing literature and emphasizing the study's contribution to the field. Although the discussion mentions the results, it does not delve into the underlying mechanisms that could explain them. No, I do not think so: what can be explored using a registry was defined in details, and a rule of thumb and practical applications were also inferred form this study and from the literature as explained in the conclusion section. There appears to be a slight bias in favor of STCTS, as negative aspects or potential drawbacks are not as thoroughly studied as positive results. Negative sides and drawbacks are provided in the discussion.

The manuscript generally provides practical insights for clinicians regarding using STCTS in specific patient populations. No, in general population: osteoarthritis is a general population. This is what it is missing in literature. Although the conclusion provides insights, it lacks specific recommendations for clinical practice or further research. I beg your pardon: there are guidelines at the end of the manuscript. Using an extensive registry data set increases the robustness and generalizability of the findings. The study provides a detailed analysis of survival rates and hazard ratios, offering valuable insights into the efficacy of STCTS in hip arthroplasty. The manuscript would benefit from more context regarding the clinical implications of the findings, mainly how they translate to patient care. I am the first providing it, see above. There is a tendency to favor positive STCTS outcomes without carefully examining potential drawbacks or adverse outcomes. There were clear disclaimers in discussion and also in the conclusions, as explained above. The study did not adequately account for potential confounding variables that could affect outcomes, such as patient comorbidities or variations in surgical techniques. It is not possible with such a design. The study focuses primarily on quantitative data, with limited exploration of patient-centered outcomes such as quality of life or satisfaction. Once again, it is not possible with such a design. Future research directions or unanswered questions should be developed to guide further research in this area. See the conclusions

Comments on the Quality of English Language

Minor grammatical errors throughout the text can be corrected to improve readability.

Revised

Reviewer 2 Report

Comments and Suggestions for Authors

Reviewer Comments

Summary

This manuscript evaluated the long-term performance of the single-taper conical tapered hip stem for treating primary osteoarthritis. A regional arthroplasty registry data was used to compare the performance with another group which included cementless stems. Various statistical methods were used to determine and compare the two groups' survival rates and hazard ratios. The procedures and results were well described, but a few aspects of the statistical methods applied in the study were missing. Please see my comments listed below.

General Comments:

1)      Table 1: P values are provided by comparing the two groups. Please add p values for the parameters considered for each group (e.g., Gender: Female vs. Male).

2)      Did you perform any statistical checks to verify if the data is normally distributed?

3)      Please elaborate on why the Welch Two sample t-test, Pearson’s Chi-squared test, Log-Rank test, and Cox multiple regression models are more suitable to apply for the data in this study instead of applying other statistical tests in their category.

4)      Figures 2, 3, and 4: Please enlarge the size to make details legible.

Comments on the Quality of English Language

Minor editing of English language is required.

Author Response

Summary

This manuscript evaluated the long-term performance of the single-taper conical tapered hip stem for treating primary osteoarthritis. A regional arthroplasty registry data was used to compare the performance with another group which included cementless stems. Various statistical methods were used to determine and compare the two groups' survival rates and hazard ratios. The procedures and results were well described, but a few aspects of the statistical methods applied in the study were missing. 

Many thanks for your precious evaluation and the time you spent.

Please see my comments listed below.

General Comments:

  • Table 1: P values are provided by comparing the two groups. Please add p values for the parameters considered for each group (e.g., Gender: Female vs. Male).

Checked. All provided. Unluckily, the layout of the table is not helpful to get it, but it is required

  • Did you perform any statistical checks to verify if the data is normally distributed?

Yes, we did. The statistical analysis was uncorrectly reported and was revised as follows.

Change: line 140 Continuous data were expressed with a median and interquartile range, as they do not follow a Gaussian distribution. The qualitative data were expressed as frequencies and percentages. Demographic and implant related data were analyzed using Wilcoxon rank sum test and Pearson’s Chi-squared test.

  • Please elaborate on why the Welch Two sample t-test, Pearson’s Chi-squared test, Log-Rank test, and Cox multiple regression models are more suitable to apply for the data in this study instead of applying other statistical tests in their category.

These are the most adopted and most suitable tests for arthroplasty registry studies. This choice was also detailed in a paper.

Ranstam J, Kärrholm J, Pulkkinen P, Mäkelä K, Espehaug B, Pedersen AB, Mehnert F, Furnes O; NARA study group. Statistical analysis of arthroplasty data. I. Introduction and background. Acta Orthop. 2011 Jun;82(3):253-7. doi: 10.3109/17453674.2011.588862. PMID: 21619499; PMCID: PMC3235301.

4)      Figures 2, 3, and 4: Please enlarge the size to make details legible.

Done. The three figures were enlarged as much as possible, revised, also provided in a zip file with improved quality.

Comments on the Quality of English Language

Minor editing of English language is required.

Revised

Reviewer 3 Report

Comments and Suggestions for Authors

The submitted manuscript is a retrospective analysis from a prospective arthroplasty registry about long-term revision rates of single-taper conical tapered stems (STCTS) in total hip arthroplasty (THA) for primary osteoarthritis, in comparison to other uncemented stems. The subject is very interested and would be very important to publish, as the literature about these stems is scarce and long-term outcomes of a rather large cohort are provided. The manuscript is well-written and well-structured. I would however recommend to break down the analysis further before recommending acceptance. The main point would be patient grouping.

 Results up to 20 years of follow-up are provided, which is extremely valuable. However, revision rates in the second decade may greatly be influenced by the type of polyethylene of the inlay. Surprisingly, no revision seems to have been performed for wear. Just revisions for loosening are indicated. Would there be an issue with data registration in the registry, no option “wear” being available? If the type of polyethylene is available in the registry, a subgroup analysis separating conventional and cross-linked polyethylenes should be made.

 Increasing the size of the control group beyond 2-3x the size of the study group does not provide much more statistical power. However, as the performance of uncemented THA may vary greatly, it would probably be better to divide the control cohort into groups of similar arthroplasties and leave out those which are exotic, short and which were associated with markedly different results from the average. Groups could be mulleroid straight stems (Polarstem, Quadra, and similar), Zweymüller designs, Spotorno designs, etc. There are enough cases in the control group by far to focus down the analysis.

 The Introduction may be shortened. Particularly the 2nd paragraph contains a lot of information that should rather be moved to the Discussion. Particularly as there is some duplication of information in the Discussion.

 I acknowledge that anteversion is used throughout the literature for both the cup and the femoral torsion, particularly in the American literature. A version is however the appropriate term only for shapes with an opening, such as the cup. As the femur has no opening but would be more of a cylindrical structure, torsion should be used instead. I would recommend correcting throughout the manuscript.

 Using mean +/- SD as data descriptors is adequate only for normally distributed scalar data. In normally distributed data, 2*SD should roughly encompass 95% of the data. If 2*SD provides impossible values, then obviously data distribution is not normal. The follow-up would be an example of data better described by median and (interquartile) range. Please check data description throughout the manuscript.

 No “s” would have to added to abbreviations. I would recommend removing it throughout the manuscript from STCTS, THA, and so on.

 Fig. 1D: The cup would have an inclination outside recommended angles. Wouldn’t the authors have a radiograph with correct orientation of the cup available? Would make a better impression.

 Fig. 3: Indication of the categories should be improved. Adding titles to each figure would probably be a better option. Numbering/lettering of each figure is also recommended to be able to provide adequate information in the legend.

 Line 216: Paper is a print medium a scientific publication may be printed on. The authors should rather refer to the study they made. Colloquial language should be avoided in scientific publications.

 Overall, the authors are to be commended for their work. I am convinced it would be a quite valuable addition to the literature. Therefore, I hope the authors will address the points raised and provide a revised version of the manuscript. Thank you for having given me the opportunity to revise this work.

Author Response

REVIEWER 3

The submitted manuscript is a retrospective analysis from a prospective arthroplasty registry about long-term revision rates of single-taper conical tapered stems (STCTS) in total hip arthroplasty (THA) for primary osteoarthritis, in comparison to other uncemented stems. The subject is very interested and would be very important to publish, as the literature about these stems is scarce and long-term outcomes of a rather large cohort are provided. The manuscript is well-written and well-structured. I would however recommend to break down the analysis further before recommending acceptance. The main point would be patient grouping.

Thank you for your time and your precious evaluations

 Results up to 20 years of follow-up are provided, which is extremely valuable. However, revision rates in the second decade may greatly be influenced by the type of polyethylene of the inlay. Surprisingly, no revision seems to have been performed for wear. Just revisions for loosening are indicated. Would there be an issue with data registration in the registry, no option “wear” being available?

Wear is a reason for revision that is detailed in the form adopted by the RIPO registry. I added wear to the table with reasons for revisions. In the first versions it was listed under other causes.

If the type of polyethylene is available in the registry, a subgroup analysis separating conventional and cross-linked polyethylenes should be made.

The type of poly was available in the registry and the distribution of bearing surfaces in the two cohorts was very different, as described in the first table. As you can see, there is large amount of COC in the control group (35%) and very different poly and heads. A stratification for bearing surfaces would be very complex and scarcely informative. Moreover, the poly is very different among different companies: in the end, we would compare Zimmer poly (STCTS cohort) to other company poly liners. Comparing poly in registries is always very complex: usually, best choice in registry is comparing a specific poly in a specific shell to all the other companies. For these reasons, I would avoid a comparison XLPE/PE in the two cohorts.

 Increasing the size of the control group beyond 2-3x the size of the study group does not provide much more statistical power. However, as the performance of uncemented THA may vary greatly, it would probably be better to divide the control cohort into groups of similar arthroplasties and leave out those which are exotic, short and which were associated with markedly different results from the average. Groups could be mulleroid straight stems (Polarstem, Quadra, and similar), Zweymüller designs, Spotorno designs, etc. There are enough cases in the control group by far to focus down the analysis.

Control group was selected as all the cementless stems-conus, with no selection at all. This is a practice that is adopted quite often in registry studies.

As this is an Italian registry, some implants may sound “exotic” to a broad audience. Moreover, a similar article analyzing our THA database with stratification according to Mont et al. was already published. In order to provide a comparison, I selected some of the most commonly implanted stems in the world which are commonly adopted also in our registry and I provided a comparison with STCTS and the single stems (mulleroid, Zweymuller, Spotorno and Wagner-like), using all-cause revisions and stem aseptic loosening as endpoints. Only HR were provided for statistical reasons.

Change: Line 168. Some of the most implanted stems from the control group were selected and compared to the STCTS cohort using a multivariate analysis: ADR Smith and Nephew (single-taper conical stem); CLS Zimmer (single-taper single wedge tapered stem); Corail Depuy (Warsaw, US: single-taper single wedge tapered stem); Polarstem Smith and Nephew (single-taper single wedge tapered stem); Sl-Plus Smith and Nephew (single-taper tapered rectangle stem); Taperloc Biomet (Warsaw, US: single-taper single wedge tapered stem). The adjusted hazard ratios for stem failure were: 2.18 (CI95%: 1.14-4.18; p=0.019) for ADR; 1.54 (CI95%: 1.04-2.26; p=0.03) for CLS; 0.99 (CI95%: 0.53-1.85; p=0.968) for Corail; 0.65 (CI95%: 0.33-1.28; p=0.216) for Polarstem; 2.31 (CI95%: 1.58-3.38; p<0.001) for Sl-Plus; 0.62 (CI95%: 0.36-1.05; p=0.073) for Taperloc. The adjusted hazard ratios for stem aseptic loosening were: 3.66 (CI95%: 1.46-9.18; p=0.006) for ADR; 1.17 (CI95%: 0.56-2.45; p=0.683) for CLS; 1.05 (CI95%: 0.36-3.05; p=0.929) for Corail; 0.54 (CI95%: 0.17-1.76; p=0.305) for Polarstem; 4.17 (CI95%: 2.19-7.97; p<0.001) for Sl-Plus; 0.57 (CI95%: 0.22-1.43; p=0.228) for Taperloc.

Also Methods and Discussion contextually revised

 The Introduction may be shortened. Particularly the 2nd paragraph contains a lot of information that should rather be moved to the Discussion. Particularly as there is some duplication of information in the Discussion

Intro was shortened, in particular in the second paragraph.

 I acknowledge that anteversion is used throughout the literature for both the cup and the femoral torsion, particularly in the American literature. A version is however the appropriate term only for shapes with an opening, such as the cup. As the femur has no opening but would be more of a cylindrical structure, torsion should be used instead. I would recommend correcting throughout the manuscript.

I agree. Done

 Using mean +/- SD as data descriptors is adequate only for normally distributed scalar data. In normally distributed data, 2*SD should roughly encompass 95% of the data. If 2*SD provides impossible values, then obviously data distribution is not normal. The follow-up would be an example of data better described by median and (interquartile) range. Please check data description throughout the manuscript.

Revised. table 1 was revised as well.

Change: line 121: The median follow-up period was 10.9 years for the index stem cohort (IQR:6.6-15.3) and 5.7 years for the control group (IQR: 2.7-10.2).

Table 1 was revised

No “s” would have to added to abbreviations. I would recommend removing it throughout the manuscript from STCTS, THA, and so on.

Done

 Fig. 1D: The cup would have an inclination outside recommended angles. Wouldn’t the authors have a radiograph with correct orientation of the cup available? Would make a better impression.

Revised

 Fig. 3: Indication of the categories should be improved. Adding titles to each figure would probably be a better option. Numbering/lettering of each figure is also recommended to be able to provide adequate information in the legend.

Lettering added and legends revised. Images enlarged and improved.

 Line 216: Paper is a print medium a scientific publication may be printed on. The authors should rather refer to the study they made. Colloquial language should be avoided in scientific publications.

Revised

 Overall, the authors are to be commended for their work. I am convinced it would be a quite valuable addition to the literature. Therefore, I hope the authors will address the points raised and provide a revised version of the manuscript. Thank you for having given me the opportunity to revise this work.

Round 2

Reviewer 2 Report

Comments and Suggestions for Authors

Thanks for addressing the feedback. Just one minor correction, please cite the article provided in your rebuttal  appropriately next to the text discussing relevant statistical methods and add it to the reference list as well. No further comments.

Comments on the Quality of English Language

Minor editing of English language is required.

Author Response

Comment 1: Thanks for addressing the feedback. Just one minor correction, please cite the article provided in your rebuttal  appropriately next to the text discussing relevant statistical methods and add it to the reference list as well. No further comments.

Done. Ref 17

change: The choice of statistical analyses was based on the guidelines by Ranstam et al. [17]. 

Comment 2: Minor editing of English language is required.

Done